# The Ambivalence of Categorization:
# How Asylum Seekers in Germany Navigate Minority

Ulrike Bialas[*]

Max Planck Institute for the Study of Religious and Ethnic Diversity, Göttingen

* bialas@mmg.mpg.de

Thursday, June 23, 2022

## Abstract

**Many of the young, unaccompanied migrants who have sought asylum in Germany since 2015 do not have documentation of their date of birth or do not even know their exact age. Given the significant legal protections afforded to unaccompanied minors, these young asylum seekers therefore sometimes claim minority. Depending on one's viewpoint, their claims may seem to evidence either the immense power of the state and its categories or, conversely, the "autonomy" of migration and migrants' ability to defy the German state's attempts to categorize them. Based on multiple years of ethnographic fieldwork with young asylum seekers in Berlin, I show the characteristics of state categorization that make state categories, including minority, at once powerful and pursuable, as well as the difficulties inherent in living as a minor. I ultimately argue that rather than state power *or* migrants' autonomy, age categorization and migrants' "appropriation" of official age categories show profound "ambivalence": while young migrants may successfully pass as minors, they cannot change the rigidity of age categorization, and while they may benefit from the legal protections of minority, they must also endure its burdens.**

.

## 1. Introduction

*When Idris' boat with 200-or-so Egyptian and East African men (and some women and children) finally reached Italian waters, one of the three captains called the Italian coast guard, who assured them they would be there within the hour. "What did you do that last hour on the boat?" I asked Idris. After eight days on an overcrowded, understocked timber fishing boat, with pants turned colorless and stiff from constant submersion in salt water, limbs weak from sitting and fasting, a mind numb from prayer, and a throat sore from vomiting, I wondered, what does a person do when rescue is suddenly near? Idris said: "Someone opened a plastic bag with disposable razors and handed them out, and most of the men— maybe 150 men—started shaving. I didn't understand why at first, but someone handed me a razor and said that we needed to look as young as possible when we got to Europe."*

What does it mean when, at the end of a perilous and exhausting journey, migrants like Idris muster their last energies to shave in order to look young? Are they resisting Europe's attempts to categorize them, or are they, on the contrary, submitting to state categories so powerful they hold sway over migrants who have not yet even gone ashore?

Most asylum seekers in Germany do not substantiate their asylum plea with personal documents, particularly ones containing a date of birth.[i] Many do not even know their date of birth, as an exact age has little legal and cultural importance in their countries of origin. In Germany, however, a date of birth is crucial. Only at age 18, for instance, can Germans smoke, drive, and carry weapons or—to name perhaps

more significant acts—marry, sign contracts, and vote. For unaccompanied asylum seekers, the stakes of their age are even higher: legal distinctions between minors and adults determine the applicability of asylum and residence laws, access to housing, healthcare, education, and youth welfare, and the assignment of legal guardians and caseworkers. This has led to the bizarre situation that although many young asylum seekers intend for their migration to Europe to be a rite of passage to adulthood,[ii] they find that it is minority that can help them attain the security and freedom they risked their lives for. They therefore sometimes try to pass as minors. The German state, for its part, is generally suspicious of asylum seekers' claims to minority and attempts to verify these through forensic and visual age exams, document authentication tests, and biographical interviews. In these protracted negotiations between the German state and young asylum seekers over whether the latter will be recognized as minors, both parties evoke similar notions of youth: a childlike appearance, documentation attesting dates of birth, and the inability to live an independent life. The question therefore stands: when migrants try to attain official minority by adopting the attributes that to the German state constitute youth, are they resisting or complying with the official category age? Do they advance their freedom as they cross official category boundaries, or do they, on the contrary, reinforce the lines they traverse?

I will first provide a brief overview of the long-standing significance of categorization in the governance of migrants. Given the crude and even damaging oversimplifications many categorizations—including minority—are based on, migrants at times try to claim certain beneficial state categories, with profound effects on their selves and personal lives. I briefly describe the legal advantages of minority before turning to two stages on which age categories play out: the pursuit of minority and life as a minor. The interactions between young asylum seekers and the German state I observed during the years of my fieldwork point to three important elements of state categorization—the weight of written records, the discretion of street-level bureaucrats, and a preference for precision and consistency over truth. I ultimately follow Mahmood's (2005) call to avoid reading everything as resistance in order to recognize forms of agency not aimed at unraveling power and to thus better understand power itself.

## 2. Migrants' Categorization: False Binaries, Durable Transformations

Categorization attempts by receiving states have significantly shaped the migration experience for a long time. Groebner details the methods through which even medieval states tried to identify the people crossing their borders, the classification schemes they applied, the danger they faced of creating an "illusory world of self-confirming registration systems and files" (2007, 193), and the public anxieties over impostors, impersonators, proxy persons, and doppelgänger who used "their appearances, testimonies, and particularly their papers to substantiate their claims" (ibid., 214). From the mid-15th century on, most social groups in Europe were obliged to carry personal identity papers, and, in the late 18th century, France prohibited the use of names other than the one with which one had been registered at birth. In the 20th century, America classified its immigrants with regard to race and ethnicity—without a clear definition of either concept and often anyway evoking both just to connote a presumed assimilability or economic usefulness (Perlmann 2020). Although explicit references to "race" were largely taboo in post-war Germany, West-German governments nonetheless used the nonsensical term "Afro-Asians" to classify groups of immigrants they imagined shared particular characteristics and were thus less integrable into the German labor market than Europeans (Schönwälder 2004). In contemporary Germany, asylum seekers are classified with regard to their age and nationality, among many other markers, using medical exams, linguistic evaluations, cell phone readouts, and interviews, among other methods. Migrant classification thus has a long global history. Its goal has increasingly been disambiguation because the ambiguity of "undecidable" strangers (Bauman 1990) is incompatible with both the legibilization efforts of modern bureaucracies (Scott 1998) and the distribution of benefits in welfare states.

But: "Where there is power, there is resistance" (Foucault 1978, 95). Unsurprisingly, migrants have sought to escape the categories meant to confine them, adopt ones that might offer protection, or evade categorization altogether, just as they initially traversed borders built to keep them physically in or out. Scholars have taken migrants' mobility despite borders as support for the "autonomy" of migration, which has in turn been criticized for romanticizing migration as well as downplaying the brutality of migration control and the suffering of migrants (Scheel 2013). Similarly, even when migrants successfully "appropriate" (Scheel 2017) categories, they must then reconcile their daily lives, real selves,

and a persistently suspicious outside world with their official category membership. McNevin describes as "ambivalence" such "claims that both resist and reinscribe the power relations associated with contemporary hierarchies of mobility" (McNevin 2013, 183). In her view, the political agency of migrants lies precisely in their remaining ambivalent toward the state's categories. This already intimates the co-constitutive relationship of power and resistance. As McNevin points out, "the dispute between the primacy of sovereign power and the primacy of human mobility"—or between compliance and resistance—"can only be engaged because a temporal distinction is first assumed between original acts (that come 'before') and effects (that come 'after')" (ibid., 193). Both perspectives, however, are reductive and mistake the entanglement of power and mobility, compliance and resistance.

Widely publicized cases of asylum seekers falsely claiming identities have increasingly led the German public and policymakers to place migrants under sweeping suspicion, confronting them with what Moynihan et al. (2022) call the "matching problem" of public administrations: the state must not only define a category but also match individuals to that category. Scholars of migration, for their part, have voiced doubts about the appropriateness and benevolence of rigid, often binary categorization in the first place. Belloni (2019), for instance, shows that her Eritrean interlocutors—despite fleeing an oppressive state, despite nearly always qualifying for asylum in Europe—are not merely driven by the untenability of life in their country of origin but also lured by the promise of life elsewhere. Hamlin (2021) similarly argues that the idea that the vulnerability of refugees is categorically distinct from that of so-called economic migrants is no more than "legal fiction." The equation of native language and citizenship is another fiction—in Germany, linguistic tests help determine where an applicant is "really" from. But many migrants, for complex biographical and geopolitical reasons, speak a native language other than the official language of their country of citizenship (Blommaert 2009). Western immigration authorities and courts rarely engage with the complicated political realities in regions like the Horn of Africa nor with the international laws that would relieve migrants of the bureaucratic failings of their home states (Campbell 2011).

One might argue that categorization regarding persecution, citizenship, and legal status are unique to migrants. Categories, however, that have become more fluid for non-migrant Western populations, socially and even sometimes legally, have also remained remarkably rigid in evaluations of asylum claims. Despite recent liberalizations in Western democracies regarding the classification of genders and sexualities (Brubaker 2016), for instance, asylum decisions are still based on binary notions of migrants as either queer or not (Akin 2017), rather than allowing for the kind of fluidity increasingly accepted for native populations. Religious affiliation and religiosity, similarly, are often treated as categorical and immutable in asylum claims based on religious persecution (Madziva 2018) despite their nuance and volatility. Analogously, DNA testing requirements for family reunification promote biological, binary definitions of relatedness that are at odds with recent liberalizations for native Europeans of what constitutes a family and with migrants' own diverse practices. The state's "endorsement of a biological concept of the family" (Heinemann and Lemke 2014, 498) corresponds to an emphasis on a biological concept of age—yet migrants themselves may harbor alternative definitions in both cases. Adolescence, too, has expanded for many Western youth, often into their 20s (Sawyer et al. 2018), while unaccompanied migrants still largely face an abrupt changeover from vulnerable minors to threatening adults on their 18th birthday.[iii]

In fact, we have known for a long time that definitions of childhood differ across time and place and that states are among the actors in a position to manipulate them. Ariès (1973) argues that the main change towards "modern" notions of childhood, youth, and adolescence in Europe took place only in the 17th century. It was then that youth became associated with weakness (thus deserving support) and innocence (deserving of forgiveness). These sentiments are the implicit basis of asylum laws that provide minors with a legal guardian and do not punish their border crossing. Adolescence, similarly, was only coined as a term in 1904 to acknowledge trends that "extended dependency beyond childhood and delayed entry into adult roles" (Crosnoe/Kirkpatrick 2011, 440). Age-related norms—both informal and legal—vary considerably, even within otherwise similar societies (Buchmann/Kriesi 2011; Juarez/Gayet 2014).

Given how consequential categorization is for migrants but how poorly the available categories often represent migrants' reality, it is hardly surprising that migrants would try to use state categorization to their advantage. Moreover, we know from other contexts that scrutiny compels people to lie. Sadiq (2008), for instance, shows how migrants from the Global South, in the absence of legitimate papers, obtain fake citizenship documents of countries whose "paper citizens" they consequently become. Young asylum seekers in Germany are under constant scrutiny by the state, whether during repeated interviews

regarding their asylum case or biannual investigations by the youth welfare office. Such frequent scrutiny not only makes you more vulnerable but also more likely to—intentionally or unintentionally—provide false information.

Dishonesty is thus not exceptional. And yet it feels uncomfortable to say that refugees—already a stigmatized population—often do in fact keep silent, embellish, twist the truth, or even fabricate it. If we acknowledge this at all, we tend to rush to affirm that their lies are "innocent" (Sayad 2004). What is actually perhaps more interesting than the fact that refugees do not always speak the truth is that fabrications which are initially purely instrumental can over time have profound effects on their personal lives and senses of self. Not only does the asylum system produce the paradoxical effect of at once silencing identities and necessitating identity *performances*, as Bohmer and Shuman (2007, 606) point out. Identity performances also at times become identities, hinting at the mutual constitution of category and self that Hacking (2006) has called "dynamic nominalism." Menjívar and Lakhani (2016) have indeed shown that the personal transformations immigrants undertake because they hope these will improve their legal status—such as volunteering, joining the army, or getting married—can eventually have genuine and long-term effects on their behaviors, outlooks, and selves that far outlast any need for papers.

Like other people who interact with public administrations, migrants thus "attempt to match themselves to state-created categories" (Moynihan et al. 2022, 1). Indeed, "where there is power, there is resistance." But Foucault goes on to say: "And yet, or rather consequently, this resistance is never in a position of exteriority in relation to power" (1978, 95). The question then remains: When young asylum seekers try to pass as minors, are they resisting power or complying with it? Or are resistance and power simply too entangled, as concepts like "ambivalence" (McNevin 2013) and "appropriation" (Scheel 2017) suggest?

## 3. Methodology & Field Site

In 2016, I began shadowing forensic medical examiners at a German hospital to learn about the age exams they were conducting on young unaccompanied asylum seekers by order of youth welfare offices and family courts. Soon after, I also started volunteering with an organization for unaccompanied minors in order to learn more about their legal receiving context. Here, I met—mostly male—asylum seekers whom I initially accompanied to appointments at the BAMF[iv], Foreigner Registration Office, Youth Welfare Office, and other public agencies. Although they all had issues with their age—most claimed to be minors but had been determined to be older—I did not know how old they "really" were, and I did not press them about this. As I spent more time with them—and they presumably realized that I did not care how old they were and would continue to support them regardless—several told me that they actually were young adults. I stopped volunteering with the organization but continued to spend time with some of the young men I had come to know. I still joined them for official appointments but also increasingly spent their free time with them. I recorded over 2,000 pages of typed field notes as well as several notebooks of handwritten notes.

Of course, I still could not be sure of their "real" ages—but neither did I care. I was not interested in their age as a chronological fact but in their pursuits of minority, their motivations and strategies, as well as the consequences of age in their lives. I needed to get comfortable in the ambiguity and indeterminacy of their identity. As MacLeod (1987) has argued, ethnographic research itself often becomes a microcosm for the topic under study, and there was no reason for me to assume that in a world where people hid their identities, I was somehow privy to an inside view.

## 4. The Advantages of Minority

In order to appreciate the high hopes young asylum seekers place in minority, we must first understand the asylum process in Germany and the many points at which an applicant's age makes a difference. Various international agreements, such as the 1990 UN Convention on the Rights of the Child (UN 1990), declare minority a protected status and mandate the special reception and treatment of unaccompanied minors. In Germany, minors do not have to file for asylum until they turn 18, as they cannot be deported, and the Dublin III Regulation does not apply to them. With time to settle down and

the support of caseworkers and guardians, minors are usually better prepared for the asylum interview than adults. Even the rejection of an asylum plea is more easily compensated by those who entered Germany as minors. They usually speak better German, have perhaps graduated high school, and are thus more likely to get an apprenticeship, which can secure their residence through §60c of German residence law. Moreover, asylum seekers who are under 21, have been in Germany for at least four years, and meet certain integration requirements may apply for a residence permit under §25a of German residence law.

Not only do residence laws distinguish resolutely between minors and adults, the day-to-day life of variously aged asylum seekers in Germany also differs greatly. Minors can enroll in regular schools, while adults are segregated in unaspiring language and integration courses. Adults live in camps with little privacy or comfort, while minors live in apartments provided by the youth welfare office, often into their 20s. Here, they are able to socialize with German teenagers, offered free-time activities and tutoring, and assigned individual caseworkers as well as a legal guardian.

The advantages of those who were once minors in Germany—a better education, networks of helpers, a well-prepared asylum interview, and legal alternatives to asylum—last into their adulthood. Such sharp legal distinctions and divergent realities between minors and adults lead many young asylum seekers to believe that minority is their only chance for a secure and fulfilling life in Europe. Those who initially deem themselves lucky to be recognized as minors, however, soon find out that minority is not a panacea and actually comes with its own distinctive challenges.

## 5. Becoming a Minor

Young migrants in Germany usually attain their official age through lengthy and taxing negotiations with state offices or through an age exam.[v] I will describe how Paul from Guinea, Samir from Sudan, and Idris from Ethiopia became minors, and how Zeinab from Afghanistan and Leandre from Cameroon were unable to settle their age. Their experiences demonstrate three characteristics of state categorization of migrants in Germany that make categories at once powerful and pursuable: the weight of written records, the discretion of street-level bureaucrats, and the prioritization of precision and consistency over truth.

### 5.1. Paul: The Weight of Written Records

Paul had been classified as 16 in a visual age exam in one city, then classified as 18 a few months later in another and thus relocated to Berlin. A social worker at Paul's camp in Berlin noticed the quiet, withdrawn young man, who looked younger than the other residents. She spoke French and began talking to him, eventually learning of his two age exams and his minority, on which he insisted. She called the organization I was volunteering with and reported that someone needed help fixing their age.

With the help of volunteers and social workers, Paul asked BAMF to give him the benefit of the doubt—given his two conflicting age exam results—and end his Dublin period, which they agreed to. Paul then went to family court with the written confirmation that he no longer "had Dublin," to quote the jargon used by asylum seekers, and that, accordingly, even BAMF had doubts about his majority age. The family court accepted this argumentation and assigned him a legal guardian. Paul returned to BAMF and informed them that the court had assigned him a legal guardian and must, therefore, be convinced of his minority. BAMF accepted his minority, and Paul was now able to cast his asylum plea anew and move to an apartment run by the youth welfare office. Different agencies—such as the family court, youth welfare office, BAMF, and Foreigner Registration Office—often have different identities on file for asylum seekers, and playing these off against each other, that is, approaching one agency with the identity recognized by another, is one way asylum seekers pursue identities. "Instead of openly contesting the rules and regulations of border regimes, practices of appropriation simulate compliance with these rules and regulations, but only to clandestinely subvert them" (Scheel 2017, 393). Paul had ostensibly complied with the order of age categorization but actually resisted his place within it. He had "appropriated" not only the state's categories but even its administrative structure.

Paul took full advantage of his new supportive environment, building close relationships with caseworkers, eagerly learning German at school, and joining a youth soccer team. As a Guinean, his chances of getting asylum were slim, but his minority could secure his future in Germany: living in a

youth welfare apartment, he was not deportable; focusing on his education could enable him to get a residence permit through an apprenticeship; and as someone who would have lived in Germany for four years before turning 21, he might eventually qualify for a residence permit.

## 5.2. Samir: The Discretion of Street-Level Bureaucrats

Becoming a minor and using the legal advantages of minority, however, is not always as smooth as in Paul's case, as Samir's odyssey illustrates. Samir first arrived in Hamburg and stayed with older friends from Sudan. They said that in order to live with them, he needed to be an adult. He also believed—mistakenly—that adults received more money from the state. Given his young looks, he even claimed to be married to make his majority age more credible. His plan backfired: As an adult, he was relocated to Berlin and put up in a former school gym with hundreds of others.

Samir soon understood the advantages of minority but had little hope that he could become a minor again. During his asylum interview, however, he mentioned to the interviewer that he was actually 17, and to both our amazement, the man changed his date of birth in the Central Register of Foreign Nationals and instructed Samir to get a new ID card from the Foreigner Registration Office.

We first sat in the Foreigner Registration Office's notoriously slow-moving waiting rooms, then decided to wait outside by the canal. Samir took his shoes off and wiggled his toes, telling me he had worn them for two days and two nights straight because he had not been home. If only he could be 17, he swore, he would turn his life around. If he didn't have to live in a camp, he would stop dealing, enroll in school, and go to bed each night by 11. He would buy a TV and spend the evenings inside, watching shows to improve his German. Maybe he would even get a gym membership.

The clerk corrected Samir's date of birth on his ID with a ballpoint pen and accidentally recorded a different birth month from the one Samir had given BAMF, making him even three months younger. Samir barely made it out of the clerk's office before jumping up and down: "Seventeen! Seventeen!" He insisted we go to the office that assigns schools immediately, determined to make good on his promise to change his life if the Foreigner Registration Office changed his age. Only a few weeks later, however, Samir declared "Fuck 17!" He had indeed enrolled in school, but his patchy German had made him difficult to place. He spoke quite well, but without formal education, he was barely literate and only knew "Street German," as he called it. Being in a beginners class was frustrating and humiliating. LAF,[vi] from which adult asylum seekers receive their money, had stopped paying him because they were no longer in charge. His camp had kicked him out because they weren't allowed to house unaccompanied minors, forcing Samir to move into a shelter for youth in crisis, with 24/7 supervision. We went to the Youth Welfare Office repeatedly reminding them of their obligation to put him up, but its employees were weary of asylum seekers suddenly being years younger. They purported to be short on suitable apartments, while making no secret of the fact that they did not believe Samir was 17. The day before his 18th birthday, the youth crisis apartment forced Samir to move out because they were not allowed to house adults. We went to a youth homeless shelter, hoping he could spend the night there, but they explained they would have to kick him out at midnight, when he turned 18. The adult homeless shelter also refused to take him because he would only be an adult after midnight and their intakes ended at 10 pm. Broke, homeless, and humiliated, Samir realized that so far, turning 17 had solved nothing. He had lost the only advantage of being an adult—his freedom—and gained little in return. The "autonomy" he had exercised initially seemed to get him what he desired—minority—but ultimately proved ineffectual against a state bureaucracy designed to stifle it.

## 5.3. Idris: Precision over Truth

One of Idris' childhood friends had come to Germany a year before him and advised Idris to say he was 16. Idris was about 22 years old at the time and thought that 16 was too much of a stretch, but he did want to be a minor, so he went to Berlin's clearing office for unaccompanied minors and said he was 17. A visual age exam confirmed his claim and he was assigned to a former hostel that had been turned into a shelter for unaccompanied minors, from which he was later able to move to an apartment provided by the youth welfare office.

Although Idris had been a minor for his entire time in Germany, in summer 2019, Germany passed a new law (*Geordnete-Rückkehr-Gesetz* or Law of Orderly Return) that obligated asylum seekers to prove their identity with a passport. In order to apply for a passport from the Ethiopian embassy, Idris asked his brother for a birth certificate. His brother went to the local citizen center in Ethiopia, where he was issued the document without providing proof of the information he had asked them to attest. Idris' brother sent the birth certificate to Berlin, but when it arrived, we saw that Idris had made a mistake. He had given his German date of birth, but Ethiopia does not use the Gregorian calendar, and so, on paper, Idris was now 11 years old. Idris was very upset, not only, as he explained to me, because of the additional hassle, but because the birth certificate that identified him as 11 was an unwelcome reminder of the root of this mess.

Idris used an online converter to figure out the Ethiopian equivalent of his European date of birth, and his brother obtained a new birth certificate. This time everything worked out: Idris showed the birth certificate to the Ethiopian embassy in Berlin and was issued a passport that matched his official German date of birth. A few days later, I wondered aloud whether BAMF would be suspicious of the alleged ability of an Ethiopian minor without formal education arriving in Germany worn out from the journey but able to convert his date of birth from the Ethiopian to the Gregorian calendar. A social worker who was with us agreed with my reasoning but said that BAMF demanded precision even when it was highly improbable, ultimately choosing precision over accuracy. Just like Paul, Idris had thus "appropriated" Germany's age categorization system, responding to its unrealistic demand for a precise date of birth in the only appropriate way—with an unrealistically precise date of birth.

## 5.4. Zeinab & Leandre: Living with Contested Ages

While Paul had successfully been able to play various state agencies against one another—consistent with McNevin's appraisal of ambivalence as migrants' "political resource, rather than a strategic handicap" (2013, 185)—the circumstance that different agencies recognize different identities can also lead to much trouble and biographical paralysis. When you are neither a child nor an adult—"ambivalent" to both categories—neither the agencies for adults nor those for minors feel responsible, leaving you in a liminal space of total ineligibility.

Zeinab's sister had fled to Germany after her husband tried to kill her. When this man also made threats against Zeinab, her brother organized a fake visa with the birth year 1992 for her to leave Afghanistan. She used this to travel to Germany, but on her Schengen visa application form, the birth year was accidentally recorded as 1993. Once in Berlin, she went to the Afghan embassy and received a passport with the birth year 2000—her self-stated age. This is also the year of birth her high school in Berlin recognized. The Berlin Senate Administration, however, ordered a forensic age exam because it considers passports from Afghanistan unreliable. This exam determined 1997 as the year of birth, which the youth welfare office then used. An employee at the Foreigner Registration Office recorded yet another year of birth, 1995, after learning from Zeinab's sister that she had been three years old when their mother died while giving birth to Zeinab. Finally, two different employees of the youth welfare office had visually estimated Zeinab's year of birth—one 2001, the other 2002.

The family court accepted Zeinab's minority, so she continued to have a guardian, even though her official ID showed her to be an adult. Zeinab's health insurance listed her as a minor so her legal guardian had to agree to medical procedures. Her school listed her as 18, but she was not allowed to go on school field trips because her ID made her too old for youth hostels. After the forensic age exam, the youth welfare office would no longer house her, but the LAF also refused her, pointing to her minority, so Zeinab had to move in with her sister. When people have uncertain ages, they often fall between the spheres of responsibility of various agencies. An uncertain identity also prevents you from participating in perhaps smaller ways that are, however, no less important to a sense of belonging and quality of life. Young men with multiple dates of birth, for example, could not join sports teams as the coaches needed clarity about whether to admit them to the youth or the men's team. When he was "between ages," Paul was unable to start psychotherapy because neither adult nor youth therapists were comfortable taking him on.

Leandre's identity was especially detrimental. His mother had left Cameroon for Germany when he was a preteen, married a German woman for papers, and eventually brought over Leandre on a fake visa. Leandre was, he said, 17 when he came to Germany, but the visa he had traveled with belonged to a 12-year-old, and his fingerprints were now permanently linked to this identity. Leandre was an exceptionally

smart, motivated young man who had taught himself intermediate German within only a few months. He wanted to study medicine or at least do an apprenticeship in nursing. Yet even though he was visibly not 12, he was not allowed to study, start an apprenticeship, or even work or do an internship as a 12-year-old. Several apprenticeship interviews had gone well, but as soon as the firm asked to make a copy of his ID, they retracted their offer as they were not allowed to employ children. He also could not enroll in a regular school, because as a 12-year-old he would have to be enrolled in sixth grade, but no sixth-grade-teacher would allow a grown man in her class. Leandre could not live with his mother because she lived in a community project exclusively for women, which allowed boys but not men on its premises, but he also was too young for youth welfare (which starts at 14), let alone a room at a camp for adults. He also could not file for asylum because as a 12-year-old, he needed a guardian to do so for him. He did not want to alert the family court to the fact that his mother was not officially his guardian lest it jeopardize her papers.

Leandre's and Zeinab's cases show the legal precariousness of those who are both minor and adult—and therefore neither. Ambivalence, if self-chosen and used strategically, indeed has the potential to be a resource to migrants (McNevin 2013), to shield them from the detriments of certain category memberships. When imposed upon them, however, ambivalence may conversely disqualify them from these same categories' benefits and protections.

## 6.    Living as a Minor

Even when migrants succeed in becoming a minor, they then face the challenge of living as one. If their minority was merely a fabrication, they fear being found out. Distrust and suspicion taint their relationships to German volunteers and caseworkers and even to their peers. They also often feel infantilized by the restrictions placed on minors and the rules in youth welfare facilities, feelings which are exacerbated by the fact that they often came to Europe precisely to become men. Their intended "rite of passage" (Turner 1967) has become a "rite of reverse passage," as Galli (2018) elegantly puts it.

### 6.1. Fear and Distrust

Many asylum seekers who have given false information feel guilty. Idris especially struggled to maintain his self-image of a good, honest person in the face of his false age claims. He would shudder with embarrassment when telling me that his caseworker called him and his best friend Yakob "my children." Idris also suspected some people knew he was not a teenager. His German teacher, herself only in her mid-20s, treated him and another student more strictly than his classmates, and he was sure this was because she had correctly guessed their age. Idris also advised me not to congratulate asylum seekers on their birthday because most, he insisted, would only be ashamed by my well wishes or think I was mocking them. Idris himself avoided seeing people on the day of his official birthday who might congratulate or give him presents.

As a devout Muslim, Idris wondered whether his false age claims had violated the Islamic dictate to tell the truth or whether they fell under the exemption that breaking rules was permissible when the believer's life was at stake. He asked a *mufti* for advice and, when the man informed him that he had committed a great sin, was very upset and even considered correcting his age. But then he thought about the implications. He would have to tell people the truth, he might not be able to continue high school, and perhaps he would have to repay his youth welfare. He went back and forth, imagining in horror that he would be stuck with this date of birth for the rest of his life but eventually decided that the consequences of trying to correct it were too severe.

Idris developed two strategies for assuaging his feelings of guilt and reestablishing his identity as essentially good: suspecting everyone else of having lied and establishing a hierarchy of more and less acceptable lies. He seemed to relish in speculations about others' identities and came up with evermore hyperbolic descriptions of the deceit around him. When I showed him official statistics about asylum seekers' origin countries, he laughed and said: "The Iraqis are from Iran, the Afghans from Pakistan, the Eritreans from Ethiopia, and the Syrians from Lebanon. Take my word for it." It seemed that he wanted to convince me that lying was not a personal shortcoming but almost inevitable. Idris also explained to

me that lying about one's age was less bad than lying about one's nationality because he, too, had been 17 once, whereas he had never been Somali or Eritrean.

The fear of being found out significantly shaped relations with German helpers. Yakob was invited to spend Christmas with Marie, a volunteer at the organization for unaccompanied minors. Marie, who suspected Yakob's real age but did not explicitly tell him, recounted that her sister (who was actually younger but officially older than Yakob) said over Christmas dinner: "Yakob, you don't look your age at all. You look much older." Yakob seemed uncomfortable and responded: "Yeah, the flight really ages you."

Fear and distrust not only shaped relations between asylum seekers and German helpers but permeated even intimate relationships. I would sometimes hear the young men speculate whether a friend or girlfriend they had argued with would betray them to other people or even the public authorities in revenge. I also noticed that asylum seekers often did not know the identities of even their closest friends and avoided giving them personal information which they might accidentally reveal to others or deliberately use against them if the friendship turned sour. Samir had sleepovers with his friend Whizz several times a week, during which they would pool their money to buy weed and then hang out at the park before going home to doze off to Netflix. After months of hanging out, Samir left Whizz and me alone one evening to get soda. While he was away, I said something to Whizz about Samir, and when Whizz quizzically replied "who?", it took me a few seconds to realize that they did not know each other's names. The next day I asked Samir about Whizz—his real name, where he was from, his age—but Samir said he had no idea and that it was none of his business. Samir saved all his friends' numbers, including mine, under invented names or even just emojis. When he lost touch with a friend and pondered how he could find him, I suggested we call camps to see if he was living there or ask the citizen center for information. Samir grinned at my naivety: "You think I know his European name or birthday or anything?" Samir and many of the other young asylum seekers I knew thus cultivated ambivalence to such an extreme extent that they did not settle into a concrete identity even around their closest friends. While this mitigated the risk of being exposed as well as, perhaps, the cognitive dissonance that can come from living with multiple identities, it also frequently undermined the potential for meaningful relationships and the kind of solidarity that might have made life in a new country with an insecure legal status easier.

## 6.2. Infantilization and Emasculation

Minority not only creates feelings of guilt and distrust but is also at odds with desires for independence and self-actualization—the hope for migration to be a rite of passage to adulthood—and therefore often experienced as infantilizing and emasculating. When Samir ranted "fuck 17!" just weeks after finally becoming a minor, I was reminded of an age examiner conference I had attended two years earlier, where a speaker had quipped that asylum seekers wanted to be "forever 17." Yes, Samir had desperately wanted to be 17, thinking of minority as a panacea. But after years of self-determination, being a minor was hard.

In youth welfare, to which he was eventually admitted, Samir lived in a shared apartment. Social workers were present on weekday afternoons, but also occasionally checked in unannounced, and Samir was expected to participate in a weekly group dinner, meetings with his caseworkers, and a weeklong group trip in the summer. He had a 10 pm curfew on school nights and could only have sleepovers on non-school days with the caseworkers' permission. His caseworkers were in close contact with his teachers and doctors. Samir was paid his youth welfare in small weekly installments, was expected to save 50 Euros a month, and contribute to communal expenses.

Initially, Samir was highly motivated and saw these rules as a chance. Soon, however, his motivation yielded to old habits. Samir narrowly escaped expulsion from youth welfare numerous times, and after ten months, he himself left. He was tired of the supervision, the scolding, and the small installments of pocket money, and returned to the camp, where he would again have to share a room but where there would be no curfews or check-ins and he would receive his monthly state allowance all at once. Despite placing great hope in minority and youth welfare, in the end, Samir was unable to overcome behaviors he had adopted throughout the years of being an adult.

Samir's issues were by no means unique. Idris thought about leaving youth welfare many times, usually after something happened that he viewed as an infringement on his privacy. Once, his caseworker told

him that neighbors had filed a noise complaint at night, and when Idris told her he hadn't even been home, she said he was required to let her know if he slept elsewhere. Another time, Idris was ill, and his caseworker called his doctor and had him explain Idris' condition. Idris was very upset about this, because he was not a minor at the time and his caseworker not legally responsible for him. Similarly, as an 18-year-old, doctors had given Paul large quantities of his antidepressants, and he often took more than the recommended dose; when he turned 17, however, his caseworkers held on to the pills and only handed them out in small portions, despite Paul's protest. Even entering youth welfare involves making yourself small. In the application and during the interview, you have to argue that you are developmentally delayed and unable to care for yourself. You have to repeat this self-debasement every six months in order to be allowed to stay. Importantly, the behaviors and personality traits that are advantageous to making it *to* Germany are a liability to making it *in* Germany. Young asylum seekers have to be self-sufficient in order to survive the journey, but in Germany this self-sufficiency was held against them by employees of the youth welfare office who argued that they surely did not need help since they had made it to Europe all on their own.

Idris and Paul often told me about friends at home who had gotten married, built homes, and become fathers. As they showed me baby pictures, they speculated how many children they would have by now if they hadn't left. Their regression into boyhood was even more stark relative to their former peers who were launching adult lives. One of the costs of portraying yourself as younger is being emasculated at a time of life when masculinity is a primary source of self-esteem. While other young men accentuate every physical sign of impending manhood and try to behave in ways that make them seem older, young asylum seekers regress into boyhood in search of protection. Importantly, this regression not only entails emotional hardship but also stymies the kind of citizenly life that could later help them secure their residence. After all, being economically self-sufficient and starting a family are integral to both scholarly theories of citizenship and legal prerequisites for residence—and yet minors are unable to do precisely such citizenly things.

Given the struggle of reconciling one's desire for adulthood with the necessity for minority and the impossibility of experiencing one's own life as permanent ambivalence, some young asylum seekers try to internalize their official age. A few months after Samir's age change, he started referring to himself as 17, even when it was just the two of us. He also told me about a woman who was into him, but said that at 23, she was too old for him. I assumed he was joking and laughed, but he added matter-of-factly that he was 17 now and no longer dating women over 20.

In fact, asylum seekers are not the only ones who often begin treating arbitrary dates of birth as biographical fact. Despite their initial suspicions one way or the other, once a date of birth—irrespective of its genesis—had been officially recorded, caseworkers, volunteers, and legal guardians also often began to treat it as natural and real. When Samir and I, for instance, ran into his caseworker soon after he had changed his date of birth and moved into a youth welfare apartment, she began reading his horoscope to him in all earnestness, explaining "I just looked up what Sagittarius is like because you were born on November 24 now, and it fits you perfectly." Reminiscent of the collective willful ignorance parodied in the folktale *The Emperor's New Clothes*, this attitude was common among German "helpers" and perhaps contributed to migrants' own internalization of originally instrumental or arbitrary dates of birth.

## 7.   Three Constituents of State Categorization

So what can migrants' pursuits of minority and lives as minors tell us about the powerful state category of age, and perhaps state categorization broadly? At least three things.

First, the discretion exercised by street-level bureaucrats greatly affects both the categorization of migrants and the actual impact categories have on their lives. Recall how Samir only became a minor because of a sympathetic asylum interviewer and how he became even younger when the clerk who issued his new ID accidentally recorded a different birth month from the one Samir had given. Later however, Samir was unable to use the minority he had attained through the discretion of street-level bureaucrats—again because of the discretion of street-level bureaucrats, as youth welfare employees simply refused him. Lipsky (1980) defines street-level bureaucrats as public workers with discretion over the dispensation of benefits, that is, the ability to use their professional judgment when making decisions. On the one hand, discretion gives power to the state and its bureaucrats, who may even see it as an

opportunity to showcase their dominance. On the other hand, it gives some power to migrants, who may simply try their luck again and again, on a different day, with a different clerk.

Second, the weight of written records promotes an adherence to previous determinations. Paul was able to be accepted as a minor by presenting documentation of having been treated as one by other agencies, such as being considered exempt from the Dublin III Regulation and being assigned a legal guardian. A document issued by one agency had such power that it could erase the documentary uncertainty of another. In Paul's case, this was to his advantage. But it can also do harm. The US government, for instance, sometimes refuses to update naturalization certificates it knows to contain inaccurate information with information it believes accurate because this would "undermine the credibility of its record keeping" (Stevens 2017, 223). Similarly, an administrator at one of Berlin's youth welfare offices explained to me that she always treated a client in accordance with the date of birth listed on his asylum seeker ID. Albeit likely exaggerating, she nonetheless made her point by insisting: "If I see a young man before me, but his official date of birth makes him three years old, he is a toddler to me." Leandre's continued treatment as a child even when he was obviously a young man attests to this. In such cases, just as in Paul's situation, the goal is not to verify the truth—which in any case is simply not possible in this context—but to maintain consistency. This also suggests a certain *kind* of power at play, not the singular force of domination but the pervasive power described by Foucault (Rabinow 1991). The power of age categorization almost seems to lie with the date of birth itself. Neatly printed on a piece of paper—or even just sloppily scribbled with a ballpoint pen—it determines people's treatment of one another and even their sense of self.

Third, state agencies privilege a certain form of knowledge over its content. It is unlikely that Idris would be able to convert dates perfectly between various calendars. Yet his birth certificate was accepted unquestioningly because it fit the mold of bureaucratic evidence. By contrast, a birth certificate Paul eventually submitted raised some doubts. It listed a slightly different first name from the one he had given in Germany, but he insisted this was his genuine birth certificate and that certain first names were considered the same in Guinea. (Their similarity was comparable to James and its diminutive Jim in English.) To give a final example, in Afghanistan, it is common for parents to make a note of their child's birth in the Quran, and some Afghans I knew tried to prove their age by submitting a picture of this note. Tellingly, such authentic evidence met more state suspicion than mere fabrications, which even in their falsehood paralleled Germany's precision when it comes to age. An incorrect but precise document had more weight than a note in the Quran that was far closer to the truth.

These three points have significant consequences. In Zeinab's case, the discretion of street-level bureaucrats ultimately led to multiple contradictory dates of birth. State workers were then able to point to whichever date would substantiate their claims to in-/action. Unlike Paul, Zeinab was unable to play agencies out against one another. Instead, the weight of written records entrapped her. Similarly, in Leandre's case, a written record turned out to be so weighty that the blatant truth—that he was not a child—was powerless against it. His experiences showcase both the weight of written records and the state's preference for consistency over truth as well as street-level bureaucrats' ability to not exercise discretion. Bureaucrats' appeals to consistency or even insistence that they could not act often struck me as almost spiteful. At one of Leandre's appointments at the Foreigner Registration Office, where he once again explained that he was not 12 and wanted to correct his age, a clerk said: "You should have thought of that before traveling on a fake visa."

The state's power thus comes not from any specific strategy—as migrants also sometimes "appropriate" (Scheel 2017) these for their own ends—but from jumping between strategies at will: privileging consistency over truth one time, then letting street-level bureaucrats exercise discretion, then simply adhering to a previous determination. When 17-year-old Samir wanted to move into a youth welfare apartment, the youth welfare employees made use of their discretion to keep him out. When he would turn 18 at midnight, they insisted that they needed to follow the rules and expel him—when surely they could have exercised discretion here as well, only this time in his favor. It is largely up to the bureaucrats of the state when to use their discretion and when to instead cite written records or bureaucratic demands for precision.

Moreover, migrants must live with their feelings of fear, guilt, and infantilization. One cannot continuously perform a self but must also live with that self—live with oneself, as it were. Indeed, for my interlocutors the "narrative tactics" (Beneduce 2015) aimed at attaining minority were often not the

greatest feat. The life that followed was: maintaining an identity day to day and facing the suspicion of others, the guilt and cognitive dissonance within oneself, and the looming possibility of a "context collapse" (Marwick/boyd 2011) that makes the segregating of different audiences with discrepant knowledges of one's identity untenable. Fear and distrust in fact impeded the creation of "social spaces below the radar of existing political structures" (Papadopoulous/Tsianos 2013, 178). If young asylum seekers succeed in internalizing their new date of birth, they have lost a piece of themselves. If their date of birth continues to feel foreign, their life becomes a perpetual state of exception. Either way, the aging practices of the state inhibit a healthy, unencumbered coming of age and complicate young migrants' emerging adulthood—a challenging enough life stage as it is.

The state's power also lies in defining, not just determining age. Migrants may escape a specific determination—by shaving or foregrounding their assistance needs—but they have no say over definition, the fact that it is a certain kind of appearance or dependency that formally defines youth. Many young migrants insisted to me that they felt young because they had not had the kind of childhood and youth which could have prepared them for adulthood in the Global North. They may have worked to support the family, cared for younger siblings, or even gotten married at a young age. But they had not, they contended, learned how to think and act independently, how to organize and lead their own lives—crucial skills for adults in the Global North. Yet unable to challenge official definitions of youth as being born before a certain date, lacking facial hair, or being unable to cook, they must feign these in order to find support for their actual vulnerabilities, which are largely irrelevant to the agencies that protect youth in the Global North. The state's power to determine an age, which is sometimes undermined by the successful efforts of migrants to affect the results of such a determination, is thus accompanied by a power to define, which is virtually absolute. As Moynihan et al. show, the state's difficulties in matching people to official categories often arise precisely from such a "mismatch between the individual's belief about their identity and the state's definition of it" (2022, 5).

It should be noted here that time as such does not have any particular meaning; time acquires meaning only contextually. Moreover, as Cohen argues, we often use quantitative measures of time because they shroud underlying disagreement. Following Sunstein, she calls such measures "incompletely theorized agreements": proxies which "ease the path toward agreement on practices in the face of disagreement about principles" (2018, 132). There is indeed no agreement over why youth matters. One person might argue for minors' special treatment on the basis of their presumed innocence, another their vulnerability, and a third their malleability. Using a number to encompass all minors—whether innocent, vulnerable, malleable or all or none of these—allows us to govern minority without having to solve disagreement over what it is a proxy for in the first place.

But proxies are imperfect tools, and migrants use the resulting loopholes. I occasionally wondered, for instance, whether my interlocutors actually did not know their age, particularly when I learned that some of their friends and relatives back home did. Even if their date of birth had not mattered in the way it now did in Germany, I thought, they might have been able to learn it had they tried. Switching back and forth between knowledge and ignorance, proof and ambiguity, regularity and irregularity is precisely what McNevin means by ambivalence. And, to understand young migrants' claims to minority, ambivalence is indeed a more fitting term than either resistance or compliance. As McNevin argues, "a focus on ambivalence makes us more attentive to political claims whose substance and effects cannot be captured on a register of subjection–agency" (2013, 197).

But what if we were to step outside politics momentarily? Why must we read migrants' claims as political and instrumentally oriented toward legal gains at all? What if they were instead spontaneous, playful, and open-ended?

## 8.    Conclusion: Beyond Instrumental Readings of Ambivalence

As Bauman argues in *Ambivalence and Modernity*, strangers may draw power from being unknown and unclassified, from not being easily integrable into bureaucratic category systems: "Their underdetermination is their potency: because they are nothing, they may be all" (1990, 146). Yet although many category boundaries have recently relaxed in Western countries (Brubaker 2016), which could indeed allow those who "are nothing" to "be all," this has not been the case for migrants, in whose protean lives rigid categories still reign. Bauman is therefore right to note that the stranger has certain

powers in finding a place in existing category systems—as Idris, Samir, and Paul did. But not only may those "be all" who "are nothing"; to be all is also to be nothing. Zeinab's seven dates of birth robbed her of a functional age altogether. Moreover, strangers have little power over the category system itself. Their indeterminacy may make their own categorization influenceable, but the categories stand—their walls perhaps penetrable but nonetheless unbudging.

As the "autonomy of migration" approach suggests, migrants are sometimes able to traverse spatial borders—or, in the case of age, temporal-categorical borders—despite their securitization, but they have no influence over what borders exist in the first place or what it is like to live in the various unequal slots of the grids they draw. Besides, there is beauty and power to "living in truth" (Havel 1986). Perhaps real autonomy would not mean crossing borders others have erected but building a life so far from such borders their dimensions cannot overshadow it.

Migrants understand how categorization works, and they navigate its lines not necessarily with the goal of exposing these lines' arbitrariness but with the hope of traversing them to their advantage. As de Certeau (1984) describes, people cannot always reject or alter rules, but they may subvert them "by using them with respect to ends and references foreign to the system they had no choice to accept" (xiii). While early industrial workers, for instance, still tried to fight against the introduction of standardized time into their workplaces, later they instead advocated merely "for a better trajectory within the new landscape. They fought, for example, for a shorter workweek and for more leisure time rather than fighting against the time clock itself" (Tavory/Eliasoph 2013, 927). Similarly, by actively seeking out a particular position within stratification systems, asylum seekers conform to the rules while not resigning to their predeterminations. The rules provide them with opportunities for being outwitted.

Certain kinds of categories are actually inherently imitable. Butler "suggests that the reiterative structure of norms serves not only to *consolidate* a particular regime of discourse/power but also provides the means for its *destabilization*" (in Mahmood 2005, 20). When people cross category boundaries, the performability of the category becomes evident—whether this was intended by the crosser or not. By imitating gender, drag queens debunk claims about its naturalness. White converts to Islam challenge ideas of Muslims as a racial group (Galonnier 2017). The successful performance of age similarly exposes this category's falsely assumed naturalness. Young asylum seekers and those who advocate for them therefore face the core dilemma of identity politics: "Fixed identity categories are both the basis for oppression and the basis for political power" (Gamson 1995). By pursuing certain identities, such as minority, migrants inadvertently reproduce oppressive categories and help uphold the power of classification, as is often the case when "the possibility of resistance to norms [is located] within the structure of power itself" (Mahmood 2001, 212). Dissidents and Soviet officials, for instance, used the same discourse and vocabulary (Oushakine 2001).

It is difficult to determine the political value of migrants' "behaviors and narratives that are often trivialized as being simple deceits" (Beneduce 2015, 561). It might in some sense even feel more comfortable to interpret migrants' claims as political because it feels so uncomfortable to suggest they are merely self-serving. The question of this paper was in fact partially inspired by concerned colleagues, who thought that if only I were to frame migrants' identity fabrications as political acts, my data would be less stigmatizing. In this vein, Fanon and Lacaton have argued that in contexts of domination, lying is an expression of "indocility": "submission […] to power […] cannot be at all confused with the acceptance of such power" (in Beneduce 2015, 562). Beneduce applies this logic to modern migration: "We could claim that lying is often the only possible reply to the hypocrisies that regulate migration, or the laws on the recognition of human rights" (ibid.). I share McNevin's caution, however, that well-meaning scholars might simply be conflating their own political agenda with migrants', "attribut[ing] an ambition to mobility and migrants that is not necessarily there" (2013, 194). Instead, in Mahmood's words, we should "detach the notion of agency from the goals of progressive politics" (2005, 14).

So what does it mean when migrants shave in order to look young? They can try to escape a particular category, but they cannot escape categorization. They can try to affect their own classification, but they have no influence over the official definition of youth. They can succeed in their pursuit of minority, but the struggle of living with a fabricated identity never ends. McNevin and Scheel might therefore propose that shaving is neither resistance nor compliance but "ambivalence" and "appropriation", respectively: because migrants cannot change the structure, they avoid committing to specific categories or use existing ones to their advantage.

Although I largely agree with McNevin and Scheel, I want to hint at yet another possible interpretation of shaving on the migration trail. Terms like "ambivalence" and "appropriation"—although they do capture part of what is going on—assume that migrants act vis-à-vis structure. Coming full circle with Mahmood's thinking, it might be at least worth considering, however, that shaving could also signify an "ambivalence" that operates outside of hegemonic norms and terms and be an "appropriation" that is not aimed primarily at legal gain. By focusing only on resistance, Mahmood argues, we overlook "dimensions of human action whose ethical and political status does not map onto the logic of repression and resistance" (2005, 14). Perhaps trying to explain shaving in terms of power and resistance is then presumptuous, as it assumes an instrumentality and deliberateness behind this odd collective ritual it might simply not possess. What if shaving instead allowed migrants to experiment and be playful? While minority is not a universal category, childhood, to a certain extent, is. Maybe by shaving, migrants claim not only vulnerability but also space to grow and transform, to become themselves or someone else entirely, to mess up and try again, to take longer and to take detours. These were indeed themes which my interlocutors associated with youth in situations that did not require them to think of their age in instrumental terms, aimed at material benefit. Perhaps they not only felt like minors because they were obviously vulnerable but because their new life in Europe held the kind of promise of transformation, experimentation, and play that legal advantages have nothing on.

## 9.   Endnotes

[i] There are several reasons for asylum pleas without documents: migrants come from countries without extensive civil registration systems and never owned documents; they lost documents en route to Europe; their documents were taken by traffickers; or they got rid of documents to conceal their identity. Asylum seekers from Afghanistan, Somalia, and Guinea—unaccompanied minors' most common countries of origin in Germany—almost never show documents (Deutscher Bundestag 2021, 26).

[ii] The fact that young men from the Global South often see migration as a rite of passage to adulthood—a matter that is well-documented for a range of regional contexts, see for example Ali 2007 and Monsutti 2007—does not refute hardships in their countries of origin, which of course also shape their decision to migrate. For a critique of the "legal fiction" that only forced migrants are vulnerable see Hamlin 2021, and for the converse critique of our tendency to overlook the desires of forced migrants see Belloni 2019.

[iii] The Eighth Book of the German Social Code (SGB VIII) does, in fact, hold the possibility for young adults to remain in youth welfare. Simply put, the state has to prove that someone between the ages of 18 and 20 does not need youth welfare, while the onus lies with the applicant between the ages of 21 and 26. In practice, however, it is often difficult to use these age brackets to capacity and nearly impossible to newly enter youth welfare after one's 18th birthday.

[iv] *Bundesamt für Migration und Flüchtlinge*, Federal Office for Migration and Refugees

[v] Although in this paper I focus on the negotiations most of my interlocutors underwent, I want to provide at least a synoptic account of the visual and forensic age exams I studied ethnographically for a year. In visual age exams, social workers estimate an asylum seeker's date of birth based on information gleaned in a biographical interview, their character and conduct as observed during this interview, and physical features such as hair growth, skin, and overall physique. Many German cities also commission forensic age exams, particularly when social workers cannot agree on a date of birth or another agency or the asylum seeker contests it. Most forensic age examiners evaluate three radiological images: an X-ray of the left hand, a panoramic X-ray of the jaw area, and a thin-layer CT of the clavicle. These images are compared to atlases of skeletal development, which show typical images of hands, jaws, and clavicles at various chronological ages as well as distributions among the reference population. Medical examiners suggest a minimum age (the age of the youngest study participant with the same skeletal development) and a probable age (the average age of study participants with the same skeletal development). A judge then assigns a new date of birth. I found age examiners—both social workers and forensic medical examiners—to be quite conflicted by the ambiguities inherent in their politically highly controversial work. The uncertainties—even arbitrariness—of age determination also stood in stark contrast to the matter-of-factness dates determined in this way later assumed in bureaucratic practice. For more on age exams, see Bialas (forthcoming).

[vi] *Landesamt für Flüchtlingsangelegenheiten*, State Office for Refugees

## Acknowledgements

I wish to thank Saskia Bonjour, Elizabeth F. Cohen, Evelyn Ersanilli, Salah Punathil, Stephen Scheel, Jagat Sohail, and Darshan Vigneswaran for very helpful comments on earlier versions of this paper.

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
