# Peer review of "Ambiguous Ages, Ambivalent Youths: How Asylum Seekers in Germany Navigate Age Categorization"

_Migration Politics_

## Round 1 · Referee Report · Anne McNevin (Referee 1) · 2022-9-10

Report

I found this paper to be full of insights, empirically rich (the individual stories capture the theoretical points so effectively), and extremely generative. From the outset, it sets a complex tone, avoiding slippage into oversimplistic either/or modes of analysis. The opening question is a great starting point: is the deployment of minority by migrants best understood as an expression of autonomy or a mode of subjection, or both? The paper makes a compelling case that it’s both at once, or better, that it’s something that doesn’t lend itself to easy binary categorization. Deploying minority status has different effects – sometimes its advantageous to migrants, sometimes not, sometimes with mixed results – and you draw on my own theorization of ambivalence to get at migrants’ own mixed feelings about minority claims as a strategy. I offer a few comments and suggestions to sharpen the analysis and argument.

The notion of ambivalence works well in the case of minority claims, but I think there are some inconsistencies with how the concept is being deployed, or at least with how I was using it in the article you cite. My sense is that ambivalence is a feeling that migrants have in relation to certain strategies they deploy. They are ambivalent, as you show, because strategies like claiming minority status may take on a momentum that moves in directions other than those intended (such as locking migrants into infantilizing controls, or affirming the stereotype of disingenuous asylum seekers), have a complex relation to ‘truth’, and as such entail personal and political compromises. I think it’s right to say that this ambivalence can be a resource, because it keeps strategic options open, indicates that migrants are not naïve about the stakes of particular strategies or the availability of perfect options, and means that migrants can respond to situations as they arise. But I was not so sure about the claim that ambivalence can be ‘chosen’ or ‘imposed’ or even ‘cultivated’. I think what can be ‘chosen’ or ‘imposed’ are the strategies themselves, not one’s feelings about them. What you show so effectively is that it’s almost impossible to say categorically that the trajectories of migrants with respect to minority claims are wholly chosen or imposed, but sit somewhere on a spectrum of autonomy/subjection. One feels ambivalent also because there is no option that is wholly of one’s choice, that one can be wholly accountable for (it may be impossible or life threatening to tell ‘the truth’). So one can cultivate an approach to that reality that somehow harnesses that ambivalence to be as politically effective as possible (or not – you make the important point that not everything is about intentional political acts), but that’s not cultivating ambivalence per se. So I would suggest a bit of careful thinking about where ambivalence, strategies, and perhaps ambiguity start and stop as reference points. Doing so would make your use of ambivalence more precise and more compelling, I think.

Implicitly, the article seems to be in conversation with the growing literature on time and migration. You cite a couple of key sources (Cohen, etc) but it may be worth a more explicit engagement with this literature, only because your contribution to it is original and significant and ought to be specified. In my reading, you show not only how the count of time involved in age categories is used as a mechanism of border control, but also how migrants reappropriate the count to work for them. The deployment of time as a political technique by migrants is undertheorized in the existing literature. Significantly, you also show how the meanings ascribed to the passage of time operate as a kind of meta-enclosure in which the migrants in question find themselves stuck. The discussion about exactly how one ‘grows up’ (rather than when), and how one can be married with children and adult in one sense, but immature in certain kinds of culturally specific life skills, and the state’s power to define what age means (p.12) is really important I think, because it gets at the limits of linear measure of time themselves, not just disputes over the count of linear time (how old someone actually is). This kind of temporal ‘meta’ power complicates the question of autonomy versus subjection. If migrants must operate within this meta frame, is their autonomy limited to changing the dates of their birth? Or do migrants also challenge this meta frame of the meaning of time itself in some way?

There were also several empirical details mentioned in passing that struck me as important, and might be theorized more explicitly with respect to the question of time/migratoin: the shame felt around birthday celebrations; the weighing up of age-lies relative to nationality-lies as a way of compensating for one’s guilt about lying; the internalization of one’s ‘official’ age and the impact of this on one’s sense of self; and the regression into boyhood. All these examples suggest that the impacts of these minority claims will be long-term (another temporal factor in maintaining ‘migrant’ status and other hierarchies), deeply felt, somehow ‘steal time’ or dictate the temporality of life transitions, and entangle migrants in further obstacles to ‘integration’ or status or just a feeling that their bureaucratic hurdles with the state can at some point be over. There could be a whole other paper involved in fleshing out these temporal dynamics – but either way, it may be worth engaging with a bit more of the time debate. Noora Lori’s “Offshore citizens” has an excellent summary of the time/migration debates in the early chaps. You can also take a look at my own “Time and the figure of the Citizen” (Journal of Culture, Politics and Society) for some literature leads.

Alternatively, the latter section of the article where you reflect on the power of the state could be tightened and shortened. I think the point you make about the virtues ascribed to consistency and precision over ‘truth’ is important. But perhaps it can be made more succinctly to leave room for the temporal discussion. Also the point about state agents using discretion to jump between strategies at will is convincing, and makes the important point that it’s the power to switch without notice that’s key, rather than which particular option any agent takes – but I found myself thinking that that’s also what migrants are doing, so the point might also add to the general argument that autonomy and subjection are both in play.

I hope this helps you flesh out the theoretical significance of what are really illuminating empirical details, and I’m looking forward to seeing the final version in print.
Anne McNevin.
  • validity: high
  • significance: high
  • originality: high
  • clarity: good
  • formatting: -
  • grammar: excellent

Author:  Ulrike Bialas  on 2022-10-04  [id 2874]

(in reply to Report 1 by Anne McNevin on 2022-09-10)

Thank you very much for your close reading of my paper and these detailed comments. After reading your review, I think that I need to distinguish more carefully between ambivalence vs ambiguity and feelings vs justifications. As you say, feelings cannot be chosen, but I do believe that in the case of mixed feelings one side or the other of this “mix” can be played up to make it easier to live with oneself, think of oneself as decent, and ultimately justify one’s actions. I should be more precise about what exactly I believe migrants can choose or cultivate and what simply is. To this end, I should also be more precise in my use of the terms ambiguity (maintaining a vague or flexible identity) and ambivalence (having mixed feelings).

I will definitely look more into the literature on time and migration, particularly Noora Lori’s and your own work. How migrants may use time as a political technique (also as opposed to just a political resource) and how debates around time and migration are not merely about determination (how old? how long? etc) but the definition and meanings of time both strike me as particularly interesting. I am trying to examine this question of whether migrants’ sole power lies in affecting the count of time/age or whether they can also help shape its meaning in other texts I am writing, but perhaps I could engage with it a bit in this paper as well. I also agree with you that the fact that a date of birth will follow you around forever has important implications for possibilities of integration, secure legal status etc. These are all really interesing points!

---

## Round 1 · Referee Report · Rebecca Hamlin (Referee 2) · 2022-9-28

Report

This paper is on an interesting and important topic, and is beautifully written and clear, with a nice early anecdote to illustrate the human stakes. The research is careful and through, and the paper is nicely grounded in the larger literature on migration governance and categorization. I honestly have no significant criticism for this rich piece, and I look forward to seeing it in print.

I only have a few suggestions for further reading that may prove useful, because I think some of the frames they highlight should be explored in more detail:

1. Jaeeun Kim. 2022 “Between Sacred Gift and Profane Exchange: Identity Craft and Relational Work in Asylum Claims-Making on Religious Grounds.” Theory and Society 51 (2): 303–33.
[in this work, the asylum seekers are embellishing (or perhaps reinvesting in) religiosity, which enhances their asylum claims. But the implications are similar to this paper, and I think it is worth drawing out parallels between these fascinating cases because of how well Kim explores her topic].

2. Grace Tran’s recently completed dissertation at University of Toronto “Laws of Love: Negotiations of Intimacy and Legitimacy At and Beyond State Borders Through Vietnamese “Marriage Fraud” Arrangements” explores the fraught nature of binaries like “real” and “fake” marriages in such a thoughtful way, that I think it would be useful to this project, including for your discussion of positionality relative to your research subjects.
3. I thought the discussion of masculinity was really interesting and I further think that the masculinity frame could be introduced earlier in the piece and developed more carefully. Are your subjects men because this tends to happen to men? Is it more significant for men? I think this could be a real contribution to the masculinity and migration literature. I assume there is rich sociological literature on migrant masculinities, particularly among youth/young adults. I don’t know this literature but it may be useful to engage. For example, a quick search brought me to this article, which could be useful: Marcus Herz (2019) ‘Becoming’ a possible threat: masculinity, culture and questioning among unaccompanied young men in Sweden, Identities, 26:4, 431-449. I also assume there is some great stuff about unaccompanied minors/youths at the US/Mexico border, who are often assumed to be gang members and thus, violent, reducing their access to youth.

Finally, one small point: in American English, we never say the word minority to refer to being underage, we only use it to refer to minority groups (often racial minorities), so the terminology in the title and early paragraphs threw me off until I realized what the piece was about. Depending on the target audience, it may be worth changing the title and/or clarifying at first mention what the word minority means in this context.
  • validity: top
  • significance: top
  • originality: top
  • clarity: top
  • formatting: excellent
  • grammar: excellent

Author:  Ulrike Bialas  on 2022-10-04  [id 2872]

(in reply to Report 2 by Rebecca Hamlin on 2022-09-28)

Thank you very much for your kind words, ideas for further reading, and other suggestions. I'm familiar with Jaeeun Kim's but not Grace Tran's work and agree that engaging with both cases could help sharpen some of the theoretical points I am trying to make. I will also look for a way to expand my discussion of masculinity in the paper. In short, yes - most young unaccompanied asylum seekers in Germany are male, young men's claims to minority are often met with more suspicion than young women's, and infantilization in youth welfare is perhaps felt more acutely by young men than young women. So masculinity is definitely an important issue in the context of unaccompanied youth migration, age determinations, and life in youth welfare as an unaccompanied minor. Finally, I will be sure to make clear from the beginning that "minority" in my paper refers to minority age and not ethnic or racial minorities.

---

## Round 1 · Referee Report · Anonymous (Referee 3) · 2022-10-5

Strengths

  • relevance of research question and topic of article (i.e. in how far are migrants' attempts to approriate legal and practical benfits of certain categories a form of resistance/ agency or rather just reifying the power of existing taxonomies of border and migration regimes?)
  • rich ethnographic fieldwork and details (for instance about feelings of guilt of migrants lying about their age and related moral economies)

Weaknesses

  • main weakness is that line of argument is not followed throughout the manuscript (see my comments to the author for more details)

Report

Dear Dr. Bialas,
I really enjoyed reading your manuscript on the “Ambivalence of Categorization. How Asylum Seekers in Germany Navigate Minority”. The piece has the potential to make an important contribution to literatures on migrant resistance/agency as well as the categorization of migrants within regimes of governance. It also contributes with genuinely new empirical insights to literature on so-called unaccompanied minors. However, I believe the argument still needs to be clarified and strengthened before your piece can be recommended for publication in Migration Politics (MigPol). When revising your manuscript, I would like to ask you address the following point in particular:

1.) main line of argument and research questions: The main line of argument remains unclear. More precisely, your line of argument seems to shift from one section to the next. The research question you introduce right at the beginning seems to suggest that your main focus concerns the question of migrant resistance/ agency in regards to state practices of categorisation: “When migrants try to attain official minority by adopting the attributes that to the German state constitute youth, are they resisting or complying with the official category age? Do they advance their freedom as they cross official category boundaries, or do they, on the contrary, reinforce the lines they traverse?” You basically argue that migrants practices do both – subvert and reify the categorical grids and taxonomies of the border and migration regime – when they try to pass as of minor age in order to appropriate all the legal and practical advantages that come with the status of being a minor. However, if resistance/migrant agency is the main concern of your paper and research question, I would also expect this to be reflected in your literature review (see point 3). I was also wondering what the section on the three characteristics of state practices of categorisation contributes to your argument on the ambivalence of migrants’ practices of resistance/ appropriation (see point 5). If your focus are migrant practices of resistance/agency this should be reflected throughout the paper. I will specify this main issues in the points below. Moreover, I would recommend that you rephrase your research question in a way that it cannot be answered with either yes or no, i.e. “In how far….”

2.) Introduction: Your introduction reads incomplete at the moment. First, I would recommend you also include the key info on the methodology/empirical data on which this pieces is based (see point 4). Second, I would recommend that you also include some information on your theoretical stance (are you for instance basing your argument on the autonomy of migration approach or a particular reading of migrant resistance/agency as ambivalent?) and that you also give readers a flavour of your argument (without revealing everything).

3.) Rewrite literature review (2nd section): As noted above, in a piece whose main question concerns the issue of how to conceptualise migrant agency/ resistance and where it starts and where it ends I would expect the literature review to also discuss relevant literature. Your literature review discusses, in contrast, only literature on categorisation. While it is of course impossible to do justice to all the work that has been written on migrant agency/ resistance I would expect that you discuss some of the most relevant debates and works of recent years, for instance that resistance may be problematic because it is an inherently reactive concept (i.e. people react/respond to practices of government/ power by resisting it), or that it is mostly thought of as external to power, just as the long-lasting debate on the structure-agency divide. Some of the works to include are for sure Maurice Stierl’s book on migrant resistance in Europe, Vicki Squire’s 2017 article in Politics, Cetta Mainwaring’s work (see her widely cited 2016 article from Migration Studies, work in the tradition of the autonomy of migration approach such as the work by Stephan Scheel already cited by you. The point is not to discuss all the work that is available on migrant agency/ resistance but to show that there seems to be a tendency in the literature that either celebrates the agency/ resistantce of migrants or (in the case on state practices of categorisation) to emphasise the power of categorical grids and taxonomies of border and migration regimes. You could then end with some of the literature that transcends these binary tendencies by emphasising the ambivalence of migrants practices of appropriation (for instance the work by Anne McNevin, Sandro Mezzadra (2011 chapter) and Stephan Scheel you already work with). The literature on passing as someone/something else probably also helps here (see for instance Ethne Luibheids (2002) ground-breaking work controlling sexuality at the border and the passing as straight. Of course there is also all the literature in passing as white or mixed by black people in context of the apartheid regimes in the US and elsewhere. All this literature shows how people try to appropriate and play with state-imposed categories and related practices of government, thus complicating any simple either resistance or compliance/domination framing of such practices. Then you move on and elaborate on and illustrate this ambivalence of migrants’ appropriation of certain categories in the following sections, using the case of minority age.

4.) separate methodology section needed? I was really wondering if you need a separate section on methodology, especially because it is so short. I would recommend to include it (maybe in even more succinct form) in the introduction. The main reason is that this section kind of hinders the flow of the argument.

5.) role of three characteristics of state categorization for argument (7th Section)? I was really puzzled why you have this section in your piece. The main focus of your paper is about migrant practices and if their attempts to pass as minors in order to appropriate the legal and practical advantages of being categorized as such can count as resistance or rather ultimately reify the taxonomies of the border regime. So it is about migrant practices and NOT state practices of categorization. Hence, your argument should always be primarily concerned with migrant practices, not state practices or categories. Of course the two are intertwined, but you focus on migrant practices and only study state practices through the lens of the former. So rather than unexpectedly introducing a new research question on page 10, either delete this section or rephrase it in a way that it helps you answering your question in how far migrant appropriation of category of the unaccompanied minor are emancipatory and can count as resistance or are nothing more than an expression of compliance that ultimately reifies the power and efficacy of the categorical grid of the border and migration regime.

6.) more thorough engagement with key literature: As noted above, you should engage more with more recent contributions to debates on migrant agency and resistance. Secondly, I would also suggest you consider existing work on how people navigate the categorisation in terms of age as well as related assessments and tests by state authorities (see for instance the work by Sabine Netz 2019 in Citizenship Studies and 2022 in Ethnos). Also the work by Jacob Lind on the paradoxical legal situation of undocumented minors might be relevant (e.g. in Childhood 2019). Furthermore, some of the contributions to the special issue on unaccompanied minors in 2019 in the Journal of Ethnic and Migration Studies might be useful. Finally, I would recommend a more thorough engagement with the work you already reference and work with. For instance, both Anne McNevin and Stephan Scheel want to explicitly overcome the structure-agency divide. This is the main impetus of their conceptions and use of ambivalence (McNevin) and appropriation (Scheel). Hence, the critique that both “assume that migrants act vis-à-vis structure” (p. 14) is just not fair in my view.

7.) proof reading/ expression: Finally, I think your piece would certainly profit from proof reading and language editing by a native speaker. This is not to say that your English is not good, but in some passages your expression and choice of words certainly reveals that English is not your first language. For instance, I would rather be cautious with writing about “claiming minority” but rather “claiming to be minor/ of minor age”. Also, people usually speak about “claims” not “please” for asylum. Both on page 1, but there are many similar instances of convoluted language throughout the manuscript. In some cases I found it difficult to understand the point you are trying to make. For example, on page 2 you write that street-level bureaucrats show “a preference for precision and consistency over truth”. Likewise, on page 7 you claim they would seem to choose “precision over accuracy”. Can you please explain how precision differs from accuracy and what the implications of said preference are for the subjects concerned by respective decisions? And in the literature on decision-making on asylum claims I rather encountered the argument that precision and consistency (of the protection story) are taken as indicators confirming the credibility of an asylum seeker i.e. the truth of their version of things. So can you maybe clarify the argument of how precision and consistency related to truth (production) in your case?

In any case I hope you find these points helpful to revise your very promising manuscript and I am already looking forward to reading a revised version and see it published in MigPol in a not too distant future!

Requested changes

See my report above

  • validity: high
  • significance: -
  • originality: top
  • clarity: good
  • formatting: excellent
  • grammar: good

Author:  Ulrike Bialas  on 2022-12-26  [id 3189]

(in reply to Report 3 on 2022-10-05)

Thank you for commenting so thoroughly on my paper. Your comments were quite different from the other two reviewers’, so I took some time to think about how best to address them while also keeping the aspects of the paper that seemed to be convincing to the other reviewers. This has been a really productive process, and I thank you for inspiring it. In particular, reflecting on your comments made me realize that the focus of the paper, in some important ways, does not fit the question I use to open the paper (whether migrants’ pursuit of minority is an expression of compliance or resistance). Rather than delete parts of the analysis that do not exactly fit the question (and therefore threw you off) but that I nonetheless believe are an important element of the overall argument I am trying to make, I have decided to open the paper with a question that is broad enough to encompass the data and analysis that follow. I did this in consultation with editor Radhika Mongia and, with her permission, am pasting her response below.

“One route for revisions would be to follow the suggestions of Reviewer 3 and focus on the issue of migrant agency (though through a reframed research question). If you select to follow this avenue, Reviewer 3 outlines a very workable plan for how to thoroughly restructure and streamline the essay. However, selecting this route has the drawback of omitting – or, at a minimum, reframing – many important aspects of the essay. Thus, another route would be to address Reviewer 3’s legitimate concerns (regarding the focus and overall coherence of the essay) by revising the research question/s as encompassing a wider, more capacious exploration better suited to the material presented in the essay. More specifically, the essay could be framed as exploring the multiple dimensions of how migrants/asylum seekers navigate regimes of age categorization, including the power of state categorization; the functioning of state bureaucracies; the challenges of life as a minor (or in liminal, uncertain age categories); issues of migrant agency (be it as compliance, resistance, or ambivalence); and the suggestive questions of time/temporality and related notions of “youth” and “adulthood” raised by the matter of age categorization. In fact, to my mind, this latter, more expansive approach implicitly informs the present essay. Indeed, it is revealing that the current title of the essay does not refer to migrant agency. It is also revealing that some of the richest ethnographic material (especially, section 6) barely makes mention of the issue of migrant agency. Thus, to my mind, there are several benefits to the latter approach. For instance: (a) the present framing and literature review on categorization would remain relevant. Moreover, exploring a more open-ended question would ensure that the different aspects of the essay do not seem like digressions and, instead, would provide the organizational scaffolding to: (b) Inquire into the practices of state bureaucracies and of the various state and volunteer agencies that deal with asylum seekers. (c) Examine the multiple strategies migrants deploy (from visual economies [e.g., shaving] to paper documentation and playing different state agencies against each other) to meet the criteria of “youth.” (d) Engage with issues of migrant agency, including the frameworks of ambivalence (McNevin) and appropriation (Scheel), while not being obliged to make this the primary focus. (I would, however, recommend drawing on more of the literature on this issue (e.g., in particular, Vicki Squire’s 2017 essay, that Reviewer 3 also suggests, as the framework she offers would be especially useful for your purposes). (e) Address dimensions of the “moral economies” (Reviewer 3) of migrants (including the nuanced discussion of “guilt” and modalities of rationalization) as they navigate the regimes of age categorization. (f) Delve more systematically into, both, the issues of time/temporality and masculinity. (g) And, finally, the argument/s you make that “while young migrants may successfully pass as minors, they cannot change the rigidity of age categorization, and while they may benefit from the legal protections of minority, they must also endure its burdens” (p.1, Abstract) can also be effectively incorporated into this revised framework/research question. Basically, in my view, the narrow focus on the question of migrant agency versus state control does not do justice to the nuance of the ethnography and the multiple issues you raise. Thus, I strongly recommend framing the essay in more capacious, open-ended terms that would allow for the broader – and richer – exploration the essay already pursues.”

To your question what my section on the three characteristics of state practices of state categorization contribute to my argument on the ambivalence of migrants’ practices of resistance/appropriation: I would say, a lot! Migrants’ ability to resist is, unsurprisingly, shaped by government workers’ actions, so I thought it was important to tease out some of the strategies that make these effective and that contribute to curtailing migrants’ ability to act. As I enumerate, this includes the discretion of street-level bureaucrats, the weight of written records, and prioritizing precision over accuracy, but—crucially—it also involves the ability to jump between different strategies, such as citing the discretion of street-level bureaucrats when flexibility serves the goals of a particular state agency but later insisting on the rules and denying room for discretion in another instance. As I said, and as Radhika Mongia also spells out, rather than remove these parts that I believe are important for my argument, I will revise my guiding question and framing to make these sections less jarring. I will rewrite the introduction accordingly (as you suggested anyway) and also move the information on my methodology and empirical data here.

Given that I will revise my question, moving away from a more narrow one about resistance and agency, I will not focus my literature review on migrant agency and resistance, although I will include a few more sources from this field, and I thank you for your recommendations. I will also address your smaller concerns, such as explaining the difference between precision and accuracy, in the next version of this paper. Once again, thank you very much for taking the time to engage with my paper and helping me realize that it is about more than I make explicit in my current question and framing.

---

## Editorial Decision

unknown